# Encouraging improvement in HPV vaccination coverage among adolescent girls in Kampala, Uganda

Lydia Patrick[1], Sabrina Bakeera-Kitaka[1], Joseph Rujumba[1], Oliver Ombeva Malande[1,2,3,4] *

1 Department of Paediatrics & Child Health, Makerere University, Kampala, Uganda, 2 Administration Department, East Africa Centre for Vaccines and Immunization (ECAVI), Kampala, Uganda, 3 Department of Public Health Pharmacy, Sefako Makgatho Health Sciences University, Pretoria, South Africa, 4 Department of Paediatrics & Child Health, Egerton University, Nakuru, Kenya

* ombevaom@gmail.com

## Abstract

### Introduction

WHO recommends vaccination against HPV for girls before sexual debut. Uganda started HPV vaccination in 2008 as pilot programs in 2 districts, followed by national roll out in 2015. Despite the availability of vaccines against human papillomavirus (HPV) in Uganda in the period covered by the study, there was reported low HPV vaccine uptake and completion especially of the second dose in Uganda; with little information available on timely completion of HPV vaccine and the associated factors in Uganda. This study was therefore done to determine the HPV vaccine dose 2 completion and describe the possible factors associated with timely HPV vaccine completion and non-completion among girls of age 9–14 years attending the adolescent clinic at Mulago hospital.

### Methods

A retrospective mixed methods study was conducted in Mulago National Referral hospital adolescent clinic. Data were mainly collected through review of charts and folders for clinic attendance by eligible girls and focus group discussions with eligible girls that completed the 2 doses of HPV vaccine on recommended/scheduled time.

### Results

Out of the 201 girls studied, 87 girls (43.3%) had timely completion of the HPV vaccination. Knowledge about HPV infection and HPV vaccine benefits, positive peer influence and healthcare worker recommendation to get vaccinated at health facility level positively influenced timely completion of HPV vaccine. Among barriers to completion of HPV vaccine identified were: inadequate information about HPV infection and HPV vaccine, concerns about HPV vaccine efficacy and safety, unclear communication with adolescents/caregivers from healthcare workers and -stock out of the HPV vaccine.

**Data Availability Statement:** All relevant data are within the manuscript and its Supporting Information files. However, any additional inquiries or information can be made formally to the School of Medicine, Research and Ethics Committee

(SOMREC), Makerere University College of Health Sciences, reference: SOMREC Ref 2018-025; Contact email: rresearch9@gmail.com.

**Funding:** The author(s) received no specific funding for this work.

**Competing interests:** The authors have declared that no competing interests exist.

## Conclusion

Timely completion of the second dose of HPV vaccine among girls attending the adolescent clinic of Mulago hospital was low (at 43.3%) but higher when compared to earlier published reports. Interventions around improved social mobilization, enhanced outreach and static vaccination approach and education of eligible girls on HPV vaccination can help increase vaccine uptake.

## Introduction

Cervical cancer has now overtaken breast cancer as the leading cause of mortality among women of child bearing age in Sub-Saharan Africa [1], with 85% of all new cervical cancer cases and 90% of all 311,000 cervical cancer deaths in 2018 being reported in low- and middle-income countries [2–9].

Globally, cervical cancer is the fourth most common cancer among women, and about 570,000 new cases were reported in 2018 [1]. Over 200 human papillomavirus (HPV) types have been genetically sequenced [1]. These have been further subdivided into several groups based on the potential of these to cause cervical cancer, (as high-risk/oncogenic types and low risk/non-oncogenic types). Of these types, 14 are oncogenic (i.e 16, 18, 31, 33, 35, 39, 45, 51, 52, 56, 58, 59, 66, 68), with type 16 and 18 being the commonest, and accounting for 70% of all cervical cancer cases globally [1]. There are non-oncogenic HPV types (6, 11, 40, 42, 43, 44, 54, 61, 70, 72 and 81), of which types 6 & 11 account for over 90% of genital warts [6]. The HPV viruses can infect the basal epithelial cells of the skin (classified as cutaneous type) while others target the feet and hands or the inner lining of tissues (the mucosal types) of HPV that infect the lining of the mouth, the throat, the respiratory tract, or the anogenital epithelium [5]. The 4-valent (4vHPV) vaccine, is currently the EPI vaccine recommended and available free of charge in Uganda and provides protection against precancerous lesions caused by HPV infection and genital warts.

Since the licensure of the HPV vaccine in 2006, the targeted HPV vaccination rates have remained suboptimal worldwide, [6, 10, 11], with rates as low as 34% for HPV- dose 2 among adolescent girls aged 13–15 years [11, 12]. The low HPV vaccine completion rates have been linked to fears of vaccine's effect on sexual behavior, low perceived risk of HPV infection, social influences and irregular preventive care among adolescents [9, 13, 14]. An important public health goal is enhancing HPV disease prevention through HPV vaccination. Despite high efficacy and an excellent safety profile, HPV vaccine completion remains low compared to other routinely recommended vaccines worldwide [1, 5, 6, 12, 14, 15].

Uganda rolled out the HPV vaccination into the national extended programme on immunization (UNEPI) in November 2015 by adopting two strategies, the school based approach and the static (with selected outreach) routine immunization services [16]. Uptake of HPV-1 in Uganda is at 73.8% while that of the second dose (HPV-2) is 17.3%, though this rate has recently improved steadily [14]. The WHO recommends a 2-dose schedule for immunization of immune-competent girls aged 9–14 years [1]. In Uganda, little is known about the factors that hinder or facilitate timely completion of the HPV vaccine series. In Uganda, the national strategy for HPV vaccine considers a girl 9–14 years fully immunized if she has received the second dose of the vaccine at least 6 months from the first dose. This study sought to identify the rate of timely completion of HPV vaccination and the factors that influence timely completion of the recommended schedule for HPV vaccination among Ugandan adolescent girls and therefore possibly recommend ways in which factors hindering timely completion can be addressed.

## Methods

### Study design, setting and population

This was a retrospective, sequential mixed method study, where the quantitative component generated information on the proportion of girls that had timely completion of the 2 doses of HPV vaccine. The qualitative component provided insights on facilitators and barriers to timely HPV vaccine completion. The study was conducted in Mulago hospital in Kampala, Uganda. Mulago hospital is a national referral hospital for Uganda and teaching hospital for Makerere University. Immunization of the adolescents with HPV vaccine is done at the Friday Adolescent Clinic, Baylor clinic and the child immunization clinic in Mulago hospital. The Friday Adolescent Clinic is a weekly service which has been operational since May 2013 and at the time of the study had over 1,020 registered clients. It is a collaborative effort between Mulago hospital and Makerere university department of pediatrics. The clinic has an academic collaboration with the Adolescent unit at Columbia University in New York, USA that offers technical support. It is staffed by two Pediatricians, a medical officer, six nurses, one records officer, 2 psychologists, 2 social scientists and up to 4 post graduate students at any one-time training in Pediatrics and Child health. The clinic attends to children aged 9–19 years and provides holistic adolescent health care. Most adolescents attending the clinic are from within Kampala and Wakiso districts and few from other districts in Uganda. All vaccinations are supplied free of charge to eligible adolescents through the Ministry of Health's EPI program. The health education talks by the nurse in the clinic are held before the adolescents are seen by the clinicians.

The target population of the study was all girls aged 9–14 years in Kampala, Uganda and included all girls aged 9–14 years attending the adolescent clinic and girls who received the first dose at-least in the last 6 months in the clinic. The study excluded girls who received the first dose outside the clinic setting. Given that the clinic saw about 326 girls aged 9–14 years from the period November 2015 to July 2017, this study reviewed and analyzed all available and accessible patient folders during the study period. Timely HPV vaccine completion was defined as having received the 2nd dose of the HPV vaccine series 6 months after the first dose, while non-completion was defined as having one dose of the HPV vaccine series.

### Study procedures

The authorization for access and review of the patient folders was sought from the ethics committee, the department of pediatrics and Mulago hospital. These folders are usually stored safely in the clinic's record room that is locked, alongside the clinic's registry book by the nursing officer in-charge. The files were screened for the period November 2015 to July 2017, with files belonging to boys being excluded, those for girls aged 9–14 years were selected and those ≥15 years excluded. Information on the participants' socio-demographic characteristics such as age, tribe, school attendance, and presence of a caregiver, physical characteristics (height, weight and BMI) and clinical characteristics was extracted from the psychosocial interview form for adolescents. The psychosocial interview form is filled during the first visit of the clinic. Data on vaccination status of the girls were captured from the files and clinic registries. Dates of the first and second doses of HPV vaccine were obtained and girls who had received at least one dose of the HPV vaccine in the clinic were included in the study.

### Quantitative data management and analysis

Data collected were coded, entered using Epi-data version 3.1, cleaned, and exported to STATA version 14.0 for analysis. Bi-variate analysis was done to establish any relationship

between participant demographic characteristics and timely completion level. Factors with a p value of 0.2 or less from the bivariate analysis were entered into the logistic regression model for determining factors independently associated with completion of HPV vaccination, with p-values less than 0.05 were considered statistically significant.

## Qualitative data management and analysis

The study population for the qualitative component comprised girls aged 9–14 years attending the adolescent clinic and their caregivers. Six FGDs of 6–7 participants each were conducted, 3 with the girls (1 FGD for those who had timely completion, 1 for non-completion and 1 mixed FGD) and 3 corresponding FGDs with their caregivers using a standardized FGD guide (See Table 1). Participants were given phone calls to come and participate in the study according to their vaccination status on separate days. They were given more information about the study and those who consented participated in the focus group discussion. The FGDs aimed at obtaining an in-depth and aggregate understanding of factors associated with HPV vaccine timely completion and non-completion. The discussions focused on awareness and impor- tance of the HPV vaccine and reasons for completion or non-completion of the vaccine dose. Three FGDs were conducted (one with girls who had timely completion of the HPV vaccine series, 1 with those who had not completed and one mixed group). The FGDs were conducted in English for girls and Luganda language for caregivers. All FGDs were audio recorded and transcribed by the research assistant and the PI cross checked the transcripts using notes taken during the discussion for completeness. The transcripts were read several times by experienced teams to generate themes and sub-themes. Data were then coded and grouped according to the themes and subthemes manually. Direct quotations from study participants were used in the presentation of study findings. Recording and transcription of all FGDs ensured that all proceedings of the discussions are captured by LP. Transcripts were checked and analyzed by a separate analyst (OM) to determine completeness, correctness and accuracy of transcription.

## Ethical considerations

Ethical approval to carry out the study was obtained from the Makerere University School of Medicine Institutional Review Board (Ref 2018–025). Separate individual written informed consent was obtained from each participant interviewed in the FGDs and parents/caregivers of all minors. Where additional information was sought or required that was not covered by any of these levels of consent, especially for extraction from patients' files and other secondary

**Table 1. Focus group discussion guide.**

| Focus group questions |
| --- |
| 1. Have you ever heard about HPV? |
| 2. What have you heard about HPV? *(probe for HPV transmission, what HPV causes, HPV prevention if not mentioned)* |
| 3. Why is vaccination against HPV done? (*Probe for importance of vaccine if not mentioned.*) *Have you heard of HPV vaccine, if yes what are your sources and what have you heard about it*? |
| 4. Are there any adolescents you know who have not initiated or completed the HPV vaccination? What are some of the reasons? *Probe for barriers to HPV vaccination–age of adolescent, parental consent, fear of side effects, inadequate information about vaccine, sexual activity/risky sexual behaviour, HCW recommendation, availability of vaccine, boarding vs day school* |
| 5. What would you say encouraged you to receive the HPV vaccine? *Probe on the motivators of HPV vaccination–HCW recommendation, gender of HCW, awareness of HPV vaccine benefits, parental wishes or peer influence, sexual activity, perceived risk of HPV infection* |
| 6. What do you think should be done to improve uptake and completion of HPV vaccine in Uganda? |
| 7. Do you have any other comment or questions you would like us to talk about in relation to HPV vaccine? |

data, the need for such consent was waived/granted by the Makerere University School of Medicine Institutional Review Board (Ref 2018–025). Proceeds of FGDs were kept confidential. The data were coded and was only accessible to the research team. Any girl found to have received only one HPV vaccine dose and whose caregiver consented to vaccination of the adolescent was referred for receipt of a second HPV vaccine dose at the adolescent clinic. None of the participants' identity was revealed, with each participant being assigned a unique identifier. All data and the information collected in this study are safely stored. The findings are presented in this manuscript and in the supporting information files provided.

## Results

A total of 714 folders of adolescents seen in the adolescent clinic, Mulago Hospital, between November 2015 and July 2017 were reviewed, out of which 388 files were excluded (given that they all belonged to boys and girls of age $\geq$ 15 years); thus, leaving a total of 326 folders. A total of 38 folders were excluded (because the girls either received the 1st HPV vaccine dose from a center different to the study adolescent clinic or had missing information critical to the study). This left a total of 288 records for girls 9–14 years who received HPV-1 vaccine. Of the 288 girls; 201(69.8%) girls completed 2 doses of HPV vaccine, while 87(30.2%) girls received only one dose of HPV vaccine. The study further established that 87/201 (43.3%) of girls had timely completion of HPV vaccine giving a cumulative early completion incidence of 43.3%, while 114/201 (56.7%) girls had delayed completion of HPV vaccine immunization. The age of the girls ranged between 9–14 years with a median of 12 years (IQR: 10–13). Majority of the participants were in school and in various distributions of classes. The rest of the socio-demographic characteristics are summarized in Table 2.

### Factors associated with timely completion of HPV-2

The factors associated with timely completion of HPV-2 at bi-variable analysis were: Tribe of the participant (IRR = 0.68, 95% CI 0.41–1.13), Mood status (IRR = 2.54, 95% CI 1.41–4.59) and Schooling status (IRR = 2.03, 95% CI 0.89–4.62) (Table 3). A girl who was a Muganda by tribe was 0.68 times less likely to have timely HPV vaccine completion compared to a Munyankole girl. A girl who reported her mood as sad was 2.54 times more likely to have timely completion of HPV vaccine than a one who reported very happy. Girls not in school were 2.03 times more likely to have timely HPV vaccine completion than girls in school.

**Multivariable analysis.** As shown in Table 4, the factors included in the multi-variable analysis were: tribe, schooling status and mood of the participant with a p-value less or equal to 0.2 at bi-variable analysis. The independent predictor of timely completion was the sad mood of the participant (IRR = 2.54, 95% CI 1.41–4.59). Girls who reported their mood status as sad were 2.54 times more likely to have timely HPV vaccine completion than girls who reported their mood as very happy, happy or neutral.

**Qualitative results and analysis.** Three FGDs were held with girls (1 with those who had completed, 1 with those who had not completed and 1 mixed). Similarly, 3 FGDs were held with corresponding caregivers of the girls. 64.7% of the caregivers that participated in the FGDs were women aged 30–50 years from Kampala and Wakiso districts. Most were married and in gainful employment. Most of the caregivers were mothers of the participants; the rest were either fathers, auntie or siblings. The key themes and sub-themes from the FGDs with the adolescent girls are summarized in Table 5.

**Factors associated with timely completion of HPV vaccine among girls 9–14 years attending the adolescent clinic at Mulago hospital.** *Knowledge about HPV*. Knowledge about HPV was generally limited among the girls with timely completion and non-completion

**Table 2. Characteristics of 288 girls included in the study (age range 9–14 years).**

| Variables | Frequency N = 288 | Percentage (%) |
|---|---|---|
| **Religion** | | |
| Catholic | 85 | 29.5 |
| Protestant | 81 | 28.1 |
| Muslim | 23 | 8.0 |
| Other* | 21 | 7.3 |
| Missing | 78 | 27.1 |
| **Tribe** | | |
| Muganda | 109 | 37.9 |
| Munyankole | 31 | 10.8 |
| Musoga | 20 | 6.9 |
| Other+ | 75 | 26.0 |
| Missing | 53 | 18.4 |
| **School status** | | |
| At school | 247 | 85.8 |
| Not in school | 3 | 1.0 |
| Missing | 38 | 13.2 |
| **Class** | | |
| P2 –P3 | 10 | 4.0 |
| P4 –P5 | 49 | 19.8 |
| P6 –P7 | 74 | 30.0 |
| S1 –S3 | 87 | 35.2 |
| Missing | 27 | 11.0 |
| **School performance** | | |
| Very good | 15 | 5.2 |
| Good | 127 | 44.1 |
| Average | 58 | 20.1 |
| Missing | 88 | 30.6 |
| **District** | | |
| Kampala | 168 | 58.3 |
| Wakiso | 96 | 33.3 |
| Other ** | 18 | 6.3 |
| Missing | 6 | 2.1 |
| **Child's guardian** | | |
| Both parents | 175 | 60.8 |
| Single parent | 31 | 10.8 |
| Other*** | 28 | 9.7 |
| Missing | 54 | 18.7 |

*includes: Born again, Jehovah's witness, SDA

**includes: Mukono, Mpigi, Kiruhura, Luwero, Mityana, Adjumani

***includes: Grandparents, aunt

+Jopadhola, Mukiga, Mugwere, Iteso, Sabiny, Madi, Alur, Mudinka, Nubian, Samia, Mutoro, Rwandese, Munyolo, Acholi, Mugisu, Karimajong, Mukonzo, Congolese

but this did not seem to affect the completion of HPV vaccination as their caregivers were aware of HPV. Interviews showed that to some extent most caregivers who had their girls complete the vaccination had knowledge about HPV. This is evident from the responses such as when asked whether they had heard about HPV they said:

**Table 3.  Bi-variate analysis of timely completion of HPV vaccination and baseline characteristics.**

| Variables | Timely completion n, (%) N = 87 | Delayed completion n, (%) N = 201 | Crude IRR (95% C.I) | P Value |
|---|---|---|---|---|
| Age | | | | |
| 9–11 | 51 (28.7) | 127 (71.3) | 1 | |
| 12–14 | 36 (32.7) | 74 (67.3) | 1.14 (0.80–1.63) | 0.46 |
| BMI Percentiles | | | | |
| <5 | 1 (16.7) | 5 (83.3) | 1 | |
| 5–85 | 58 (29.7) | 137 (70.3) | 1.78 (0.29–10.86) | 0.53 |
| >85 | 21 (33.3) | 42 (66.7) | 2.0 (0.32–12.42) | 0.46 |
| Religion | | | | |
| Catholic | 31 (36.5) | 54 (63.5) | 1 | |
| Protestant | 29 (35.8) | 52 (64.2) | 0.99 (0.65–1.47) | 0.93 |
| Muslim | 6 (26.1) | 17 (73.9) | 0.72(0.34–1.51) | 0.38 |
| Other | 8 (38.1) | 13 (61.9) | 1.04(0.56–1.93) | 0.89 |
| **Tribe** | | | | |
| **Muyankole** | **13(41.9)** | **18(58.1)** | **1** | |
| **Muganda** | **31(28.4)** | **78(71.6)** | **0.68(0.41–1.13)** | **0.14** |
| **Musoga** | **6(30.0)** | **14(70.0)** | **0.72(0.33–1.57)** | **0.41** |
| **Other** | **30(40.0)** | **45(60.0)** | **0.95(0.58–1.57)** | **0.85** |
| **School performance** | | | | |
| **Average** | **23(39.7)** | **35(60.3)** | **1** | |
| **Good** | **39(30.7)** | **88(69.3)** | **0.77(0.51–1.17)** | **0.22** |
| **Very good** | **7(46.7)** | **8(53.3)** | **1.18 (0.63–2.21)** | **0.61** |
| District | | | | |
| Kampala | 50(29.8) | 118(70.2) | 1 | |
| Wakiso | 30(31.3) | 66(68.7) | 1.05 (0.72–1.53) | 0.8 |
| Other | 6(33.3) | 12(66.7) | 1.12 (0.56–2.24) | 0.75 |
| Child's guardian | | | | |
| Both parents | 59(33.7) | 116(66.3) | 1 | |
| Single parent | 11(35.5) | 20(64.5) | 1.05 (0.63–1.77) | 0.85 |
| Other | 6(21.4) | 22(78.6) | 0.64 (0.30–1.33) | 0.23 |
| **Mood status** | | | | |
| **Very happy** | **17(31.5)** | **37(68.5)** | **1** | |
| **Happy** | **38(34.5)** | **72(65.5)** | **1.10 (0.68–1.76)** | **0.70** |
| **Neutral** | **10(38.5)** | **16(61.5)** | **1.22 (0.65–2.29)** | **0.53** |
| Sad | 4(80.0) | 1(20.0) | 2.54 (1.41–4.59) | 0.002 |
| Liking Your body | | | | |
| No | 1(12.5) | 7(87.5) | 1 | |
| Yes | 71(36.8) | 122(63.2) | 2.94 (0.46–18.67) | 0.25 |
| Schooling status | | | | |
| Schooling | 81(32.8) | 166(67.2) | 1 | |
| Not schooling | 2(66.7) | 1(33.3) | 2.03 (0.89–4.62) | 0.09 |
| Ever had sex | | | | |
| Yes | 2(50.0) | 2(50.0) | 1 | |
| No | 65(34.9) | 121(65.1) | 0.70 (0.26–1.90) | 0.48 |

*For missing data no analysis was done

**Table 4. Multivariate analysis.**

| Characteristics | Timely completion | Delayed completion | Crude IRR (95% C.I) | Adjusted IRR (95%CI) | P Value |
|---|---|---|---|---|---|
| | n (%) | n (%) | | | |
| **Tribe** | | | | | |
| Muyankole | 13(41.9) | 18(58.1) | 1 | 1 | |
| Muganda | 31(28.4) | 78(71.6) | 0.68(0.41–1.13) | 0.74(0.43–1.27) | 0.27 |
| Musoga | 6(30.0) | 14(70.0) | 0.72(0.33–1.57) | 0.71(0.34–1.52) | 0.38 |
| Other | 30(40.0) | 45(60.0) | 0.95(0.58–1.57) | 0.92(0.54–1.55) | 0.75 |
| **Mood status** | | | | | |
| Very happy | 17(31.5) | 37(68.5) | 1 | 1 | |
| Happy | 38(34.5) | 72(65.5) | 1.10(0.68–1.76) | 1.10(0.68–1.76) | 0.70 |
| Neutral | 10(38.5) | 16(61.5) | 1.22(0.65–2.29) | 1.22(0.65–2.29) | 0.53 |
| Sad | 4(80.0) | 1(20.0) | 2.54(1.41–4.59) | 2.54(1.41–4.59) | 0.002 |
| **Schooling status** | | | | | |
| Schooling | 81(32.8) | 166(67.2) | 1 | 1 | |
| Not schooling | 2(66.7) | 1(33.3) | 2.03(0.89–4.62) | 1.68(0.88–3.23) | 0.12 |

"*This is the virus which must be prevented more especially in young children that is why we immunize them because they are vulnerable.*" (FGD 1, Caregivers of girls who completed vaccination)

Most caregivers knew it was a virus that can be prevented through vaccination and when asked how it is transmitted most answered that it was through sexual contact:

"*It is caused through sexual intercourse so that is why the government is targeting young girls to protect them*" (FGD 1, caregivers of girls who completed vaccination).

The above quotes are an indication that indeed those care takers who had their girls vaccinated in a timely manner knew that HPV was preventable.

*Knowledge of vaccine benefit among adolescents and caregivers.* Findings from the FGDs showed knowing the benefits of the HPV vaccine among adolescents especially protecting them from cervical cancer motivated them to get the vaccine as one of them explained:

"*It (HPV vaccine) protects us from diseases (HPV infections) and my mother told me that the virus has killed many people...*" (FGD 1, girls who completed vaccination timely)

**Table 5. Key themes factors influencing timely completion of HPV vaccination among girls 9–14 years attending the adolescent clinic Mulago Hospital.**

| Major Theme | Sub-theme |
|---|---|
| Factors associated with timely completion of HPV vaccination | • Knowledge of HPV<br>• Awareness/ knowledge of vaccine benefits<br>• Peer influence<br>• Health seeking behaviour<br>• Healthcare worker recommendation |
| Factors associated with non-completion of HPV vaccination | • Concerns about HPV vaccine<br>• Inadequate information about the vaccine<br>• Unclear communication from healthcare worker<br>• Availability of the vaccine |

Data were analyzed under two sets of themes; (1) factors associated with timely completion of HPV vaccine among girls 9–14 years and (2) factors associated with non-completion of HPV vaccine among girls 9–14 years.

The same was reflected on the part of the parents whereby when asked what motivated them to bring their girls to complete HPV vaccination; most agreed that they knew about the benefits of the vaccine.

Caregivers who knew that HPV vaccine prevents the girls from cervical cancer when given before start of sexual debut encouraged their daughters to have timely vaccination as one mentioned:

*"For me I know that it prevents cervical cancer in young girls who have not yet started having sexual intercourse. . ." (FGD 1, Caregivers of girls who completed vaccination timely)*

Another care taker added:

*"I have seen the number of people who have died of cancer so if you see that there is an opportunity for your child or person to be vaccinated it encourages us so you say let me give it a try so that if it works well and good but if it doesn't too bad rather than not getting it and then you regret later. And even if you don't vaccinate your child still in the future they would die of another thing so at least you try the vaccine." (Caregiver of girl who completed vaccination, FGD 1)*

In addition, caregivers who knew the recommended interval between the 2 doses of the HPV vaccine encouraged timely vaccination.

*". . . . .the dosage must be within six months. There is a bridging period which goes up to one year or otherwise you restart the dosage again." (FGD 1, Caregivers of girls who completed vaccination timely).*

The above voice indicates that knowing the interval between the first and second dose of the vaccine as well as the fear to start again once the vaccination interval was not adhered to motivated caregivers to ensure that their adolescents had timely completion of the HPV vaccination.

*Peer influence.* Findings from the FGDs revealed that some of the adolescents who had completed the dosage on time were due to positive peer influence including their siblings. The implication here is that the adolescent girls are to some extent susceptible to their influencing peers to change their attitude, values or behavior including on HPV-2 vaccine completion. When asked how they heard about the HPV vaccine and what encouraged them to come for the vaccine, most were of the view that it was their friends. One said:

*"I heard it from my friends at school." (FGD 3, girls mixed group)* Most adolescents who had completed vaccination noted that they would encourage others to go for the same as one said:

*"We shall encourage others to come and be vaccinated" (FGD 1, girl who completed vaccination,)*

The role of peer influence was also captured among caregiver participants where they said they would encourage others to take up the vaccine for their daughters as well. One commented:

*"The other thing is that since for us we have understood it well and we have your contact numbers, we shall go and talk to young girls and explain to them so for those who will have accepted we shall give them these numbers to call the doctors so that the doctors can talk to them and they come for the vaccine" (FGD 1, caregivers of girls who completed vaccination)*

*Health seeking behaviors.* It is known that adolescents not only girls generally have poor health seeking behavior and this can play a role in affecting vaccination uptake and hence timely completion. When adolescents are well they are unlikely to visit health facilities. Findings in this study reveal that part of the reasons why the targeted participants completed their dosage timely was because of their health seeking behavior and that of their peers and siblings as one of them explained:

"*For purposes of good health and my sister encouraged me because for her she always takes up all vaccines and she is healthy.*" *(FGD 1, Girls who completed vaccination timely).*

Others were of the view that they wanted to be protected from cervical cancer. Another girl mentioned:

"*I didn't want to get cervical cancer but also my friend encouraged me.*" *(FGD 3, girls mixed group)*

Caregivers who perceived HPV vaccine as beneficial to the girls were more likely to vaccinate them on time. When asked what motivated them to bring in their girls one mentioned:
"*I have witnessed many children die of cancer so I do not want my child to go through the same. Prevention is better than cure.* "*(FGD 3, caregivers mixed group)*
This pointed towards the inherent health seeking behavior by the adolescent girls who completed their HPV-2 dosage. This, however did not appear as much in the responses hence can therefore be said to be of moderate effect towards timely vaccine dosage completion.

*Health care worker recommendation.* The FGDs revealed that healthcare worker recommendation played a major role in the timely completion of the HPV vaccine among the girls. Most of the girls mentioned that they were informed about the HPV vaccine and the need to receive it in a timely manner by HCWs.

The same perspective was also reflected in the caregivers' FGDs when asked how they got to learn about the vaccine with most mentioning HWs as some caregivers explained:

"*We have a family friend who also works in this hospital as a doctor. She is the one who told us about the HPV vaccine*" *(FGD 1, caregivers of girls who completed vaccination timely)*

"*We were at the child's school in a meeting in Kitante primary school and there is a health worker who has a child there so she told us about it (HPV vaccination) and urged parents to bring their children for vaccination*". *(FGD 1, Caregivers of girls who completed vaccination timely)*

This showed that being in close contact with a HCW whether as a relative or friend greatly played a role in timely completion of the HPV vaccine.

"*For me our family friend who is a health worker here (Adolescent clinic) invited us and even promised to take us direct to get the vaccine without waiting for long because sometimes we don't have time to come and move from place to place looking for the vaccine but since she invested her time in helping us, we decided to come*". *(FGD 1, Caregivers of girls who completed vaccination timely).*

This was an indication that indeed the HCWs at the clinic attempted to create awareness about the need to get the HPV vaccination among the girls.

**Factors associated with non-completion of HPV vaccine among girls 9–14 years attending the adolescent clinic at Mulago hospital.** *Concerns about the vaccine*. The caregivers raised concerns about the vaccine as shown through their responses. They were scared that the vaccines could have expired and harm their daughters. When asked if they knew of girls not vaccinated or had only received a single dose and why. One of caregivers responded that:

"*Some when you tell them to go and get their children vaccinated, they say that the government vaccines are expired. Even if you try to talk to them and explain that government can't . . . harm its people, they say that the government waits when the drugs are about to expire and then they come up with such programs of injecting us drugs so they say they don't want to vaccinate their children. Since most of these children are young, they have to depend on an older person to bring them for the vaccine" (FGD 1, caregivers of girls who completed vaccination).*

Another caregiver was of the opinion that there is mistrust between the key stakeholders and the caregivers resulting in misconception about the vaccine.

"*The other thing is that when government programs come, some people get to know about them but others don't but even those who will have got to know about them there are things which happen in this country and they make people loose trust in the government. Like you see when they said that they had brought vaccines for hepatitis and then we went and got them but later they told us that the vaccines were fake. So now we have that fear that when we hear government programs like these we are reluctant to go" (FGD 2, caregivers of girls who did not complete vaccination).*

Such responses are indicative of a section of caregivers that do not have confidence in government programmes. Lack of confidence and trust in government programmes plays a huge role in limiting completion of HPV-2 vaccination. Such distrust towards the government programmes in the long run can be detrimental to the wellbeing of the population and access to healthcare.

*Inadequate information about the vaccine*. Inadequate information about the vaccine was an inhibiting factor towards HPV 2 vaccine completion. For instance, when asked what hinders parents from taking girls to be vaccinated some responded that not having enough information about the vaccine:

"*I think its ignorance, if someone doesn't know something they can't know its importance or the dangers of that thing so they stay there not knowing until the problem occurs to them or they see someone close to them and then they say oh I wish I knew so I request that this program (HPV Vaccination) should reach out to people and be able to explain to them what this whole thing is about." (Caregiver of girl who completed vaccination, FGD 1)*

Some parents thought that the vaccine can introduce the virus into the bodies of the girls. Others perceived the vaccine as a disguise by the government to control the population through family planning.

Caregivers mentioned that inadequate information often resulted in misconceptions about the vaccine and hindered its completion as one of them explained:

"*. . .the other thing is that the information they give us is not adequate we don't know if this vaccine is genuine if it is tested and whether it has side effects or not because we have heard*

*rumors that the government has a program to prevent child birth so if we hear that this vaccine is targeting young girls and they haven't given us adequate or detailed information so we start to suspect that this vaccine could be one of them and that is why people don't want to come*" (FGD 2, caregivers of girls who did not complete vaccination).

The foregoing is an indication of a poorly sensitized population on HPV vaccine and its importance. The implication here is that adequate education on such a beneficial health care programme is lacking and if at all it exists then it is not comprehensive.

*Unclear communication with child/caregiver from HCWs.* Unclear communication between HCWs and adolescents or caregivers hindered timely HPV vaccine completion. When asked why their children never came for a second dose of HPV, some of the caregivers explained that there was no communication regarding the need to bring them for the second dose from the health workers and hence most of them were unaware. One said that:

"*They (girls) did not get proper information from the hospital when to come back so they lost track*" (FGD 2, caregivers of girls who did not complete vaccination)

Another also said that lack of clear communication on when to return for the second dose misled them:

"*The time interval since it is long for six months, most of them (girls) go back to school and time elapses when they are still there so when they come home, they think time has expired and they miss out thinking its late but doctors should write it clearly and indicate the return date not just writing after six months*" (FGD 2, caregivers of girls who did not complete vaccination).

The above voices indicate that to some extent there was some communication breakdown between the health care system, girls and their caregivers regarding when the second dose was to be received. The assumption by HCWs that caregivers would remember when the girls were due to receive the second dose was unrealistic and led to non-completion of HPV vaccination.

*Availability of the vaccine.* One of the major health system factors that contributed to non-completion of HPV vaccine by the girls was limited availability of the vaccine. Most caregivers mentioned that the vaccine could only be found at selected centers which made it inaccessible to some girls. Others were not aware where they could access the vaccine. For instance, one said that:

"*The service is not in many places because as for me, I thought the service is only in Mulago, even if they are informed, they do not know where to get the vaccine*" (FGD 2, caregivers of girls who did not complete vaccination).

Some girls mentioned that lack of the vaccine or not being able to receive the vaccine from a nearby setting hindered their return for the second dose.

"*. . .I used to come with her (sister) but the last time we came they said that the vaccine was over*" (FGD 2, girls who did not complete vaccination).

The above voices indicate that HPV vaccination is not always available in adequate stock levels.

## Discussion

This study was carried out 2 years after the national rollout of HPV vaccination programme in Uganda, and found a completion level of the second dose of HPV vaccine within 6 months of the first dose at 43.3%. Despite a steady vaccine supply, the vaccine completion rates were low, though when compared to many reports for the same period around the world, these rates were on average higher, even when compared to an earlier national survey in Uganda that had put completion at 73.8% (HPV dose 1) and 17.3% for dose 2. While this study's reported rate was higher than the prevailing average at the time, this figure still fell below the recommended national average of vaccinating over 80% of eligible girls[1, 6, 11, 12, 15, 17]. A much more recent USA study found that 66% of all the 13–17-year-old adolescents had initiated the HPV vaccine while only 51.1% of these had received all recommended doses, figures that were way below the Healthy People 2020 benchmark set at of 80% [18, 19].

This study, as has been shown by other studies, found that knowledge about the vaccine, positive peer influence and promotive health worker education and recommendation regarding the vaccine were identified as promoters of vaccination uptake [18, 20–23]. The study likewise, as has been reported elsewhere found some factors that are barriers to higher vaccine uptakes including prevailing community concerns about vaccine safety and adverse events following immunization, vaccine supply stock outs and inadequate information about vaccines [8, 10, 12, 14, 23]. Sarah et al., in their online survey conducted among 1,653 parents, found that only 28% to 54% were ready to send their children for the second dose; blaming this low uptake to child's fear of needles, the lack of awareness about additional doses and concerns regarding vaccine safety [12]. These findings demonstrate the need to encourage the adoption of strategies addressing needle fears, utilize reminders for parents about subsequent doses, and emphasize recent HPV vaccine safety data in discussions with parents.

In the Uganda case discussed here, the fact that the girls may have been in school and thus failed to attend the Friday clinic for HPV vaccine is plausible, thus contributing to the comparable Uganda national completion levels for 2016, HPV-1 and HPV-2 was at 83% and 22% respectively while in 2017 HPV-1 was at 72% and 28% [16, 24]. This problem has been addressed partly through a shift to blending different strategies to HPV vaccine delivery where both school-based and hospital/age-based strategies are used. Most facilities do not have a clear follow up strategy for those who got vaccine dose 1 to be followed up and actively reminded and encouraged when their second dose is due so that they come for the vaccine. This means that the vaccines kept for the second dose won't be adequately used, and may expire in the facilities. Besides, the girls will not fully benefit from the protection intended from the vaccination series unless they complete their vaccination schedule. In some situations, where a criterion exists, most adolescents report having inadequate information and guidance on eligibility criteria [25–27]. These gaps need to be addressed to strengthen the HPV vaccination program.

There are concerns that this study being a fairly urban study in Kampala could have been skewed in the fairly higher finding of dose 2 HPV completion, since generally poor completion rates worldwide tend to occur more on rural than urban settings. Barriers to effective uptake of HPV vaccines in rural areas include the lack of adequate primary care centres, lack of provider recommendations for screening, lack of preventive care services in rural areas, financial and access-related limitations in rural communities geographic terrain hindering ease of transport and movement (including vaccine delivery), disparities pertaining to knowledge about HPV infection and vaccines, cultural/religious beliefs and misconceptions that have a negative impact on HPV vaccination [14, 18, 25, 28, 29].

The finding of this study that adolescents who rated their mood as sad were likely to have timely vaccination than those who rated it as very happy, happy or neutral requires further investigation. This finding is weak, could have been affected by the subjective nature of the question and difficulties in assessing mood, and unreliability of such responses in adolescents, and would require exploration in other studies that have bigger sample size, and prospective in nature or in the context of a clinical trial. This is additionally because retrospective reviews are limited to the extent to which detailed information can be obtained from patient folders. The other factors identified in this study, and that are similar to those described in other Ugandan studies or elsewhere, and that contribute to poor completion of HPV vaccination include a general lack of trust in immunization programmes, the general belief that the vaccine is harmful to the child's health, the fear of vaccine side effects, knowledge gaps or inadequate information/ communication with the child/caregiver, and unreliable availability of the vaccine (and schedule) and a lack of sensitization and support by healthcare workers and other stakeholders to create adequate awareness to the community about HPV vaccine [4, 13, 14, 17, 30–32].

## Study strengths and limitations

This is the first known study that has determined the level of HPV vaccination timely completion and the factors associated with timely completion and non-completion in a health-facility based approach in Uganda following its introduction. Additionally, this study was done in one of the only adolescent friendly clinics in Kampala and is therefore representative of an ideal adolescent clinic. Another strength in this study was that the qualitative component was included to get more information and in-depth understanding of the factors associated with HPV vaccine completion and non-completion. Also, only female research assistants were used to help ensure that the girls were free and comfortable during the FGDs.

During abstraction of information from files, some data were incomplete resulting in information or misclassification bias. The sample size was not powered to evaluate the factors associated with timely completion especially for the quantitative component. There is need for broader better powered studies, preferably from both rural and mixed rural-urban settings, with larger sample size, and also targeting not only adolescents but caregivers, teachers, and health workers, to provide a better balanced and holistic view regarding HPV vaccine uptake, and completion of vaccination. The study population therefore might be affected by selection bias and generalizability of results may thus be limited. Information bias and the small sample size could have affected the accuracy of the results. Findings of this study captured data within the clinic and some girls could have completed the vaccine dose outside the clinic. The small sample size with most study participants being from Kampala and Wakiso districts may pose challenges in generalization of the results to the entire population, but still provides useful baseline information that can be used to strengthen the program. In addition, the study did not include perspectives of boys as this group is currently not provided HPV vaccination in Uganda. Another limitation was the time at which the study was conducted, most girls had gone back to school, the fears of kidnapping that was ongoing in the country and the claims of the unsafe Hepatitis B vaccine greatly affected the qualitative component response.

## Conclusions

The timely completion of HPV–2 vaccine among girls 9–14 years attending the adolescent clinic was low at (43.3%) but higher when compared to studies done in developed countries and national completion levels in Uganda. Factors associated with timely completion of HPV vaccine were having knowledge about vaccine benefits, healthcare worker recommendation, health seeking behavior and peer influence. Barriers to completion of HPV vaccine included

inadequate information about HPV and HPV vaccine, concerns about vaccine efficacy and safety, unclear communication with adolescent/caregiver from healthcare workers and lack of availability of vaccine. Interventions around improved social mobilization, enhanced outreach and static vaccination approach and education of eligible girls on HPV vaccination can help increase vaccine uptake.

## Supporting information

**S1 Appendix. Study profile.**
(TIFF)

**S2 Appendix. Clinical characteristics of study participants.**
(DOCX)

**S3 Appendix. Data collection tools for girls and caregivers.**
(DOCX)

**S4 Appendix. Additional data collection (and FGD) tools.**
(DOCX)

## Acknowledgments

All participants in this study (Health care workers, adolescent girls and their caregivers, members of the Paediatrics department Makerere University and staff of the adolescent clinic of Mulago Hospital.

## Author Contributions

**Conceptualization:** Lydia Patrick, Sabrina Bakeera-Kitaka, Joseph Rujumba, Oliver Ombeva Malande.

**Data curation:** Lydia Patrick, Sabrina Bakeera-Kitaka, Joseph Rujumba, Oliver Ombeva Malande.

**Formal analysis:** Lydia Patrick, Sabrina Bakeera-Kitaka, Joseph Rujumba, Oliver Ombeva Malande.

**Funding acquisition:** Lydia Patrick, Sabrina Bakeera-Kitaka, Joseph Rujumba, Oliver Ombeva Malande.

**Investigation:** Lydia Patrick, Sabrina Bakeera-Kitaka, Joseph Rujumba, Oliver Ombeva Malande.

**Methodology:** Lydia Patrick, Sabrina Bakeera-Kitaka, Joseph Rujumba, Oliver Ombeva Malande.

**Project administration:** Lydia Patrick, Sabrina Bakeera-Kitaka, Joseph Rujumba, Oliver Ombeva Malande.

**Resources:** Lydia Patrick, Sabrina Bakeera-Kitaka, Joseph Rujumba, Oliver Ombeva Malande.

**Software:** Lydia Patrick, Sabrina Bakeera-Kitaka, Joseph Rujumba, Oliver Ombeva Malande.

**Supervision:** Lydia Patrick, Sabrina Bakeera-Kitaka, Joseph Rujumba, Oliver Ombeva Malande.

**Validation:** Lydia Patrick, Sabrina Bakeera-Kitaka, Joseph Rujumba, Oliver Ombeva Malande.

**Visualization:** Lydia Patrick, Sabrina Bakeera-Kitaka, Joseph Rujumba, Oliver Ombeva Malande.

**Writing – original draft:** Lydia Patrick, Sabrina Bakeera-Kitaka, Joseph Rujumba, Oliver Ombeva Malande.

**Writing – review & editing:** Lydia Patrick, Sabrina Bakeera-Kitaka, Joseph Rujumba, Oliver Ombeva Malande.

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
