## [Decision Letter · Decision Letter 0]

21 Jul 2021

PONE-D-21-09701

Encouraging improvement in HPV vaccination coverage among adolescent girls in Kampala, Uganda

PLOS ONE

Dear Dr. Oliver Ombeva Malande,

Thank you for submitting your manuscript to PLOS ONE. After careful consideration, we feel that it has merit but does not fully meet PLOS ONE’s publication criteria as it currently stands. Therefore, we invite you to submit a revised version of the manuscript that addresses the points raised during the review process.

While I think this is an important topic that warrants investigation, there were several issues with the manuscript that are significant enough that they undermine the contributions of the study. Together with both reviewers I have a number of reservations about this paper. They are outlined below.

Specifically, the Abstract should be re-arranged, the Introduction section needs to be rephrased, provided with more references to support the need to assess HPV vaccine completion and understand its determinants; cited references should be updated. The rationale of the study should be framed. The methodology of the study should be described more thoroughly and the Results section should be re-arranged according to the reviewers’ suggestions. The discussion is mostly a list of other papers and does not have a clear synthesis of the prior data; this should be corrected. Furthermore, the writing, i.e. spelling and syntax should be proofread.

We look forward to receiving your revised manuscript.

Kind regards,

Prof. Maria Gańczak

Academic Editor

PLOS ONE

Journal Requirements:

3. We note you have included a table to which you do not refer in the text of your manuscript. Please ensure that you refer to Table 3 and 5 in your text; if accepted, production will need this reference to link the reader to the Table.

Reviewers' comments:

Reviewer's Responses to Questions

**Comments to the Author**

1. Is the manuscript technically sound, and do the data support the conclusions?

Reviewer #1: Yes

Reviewer #2: No

2. Has the statistical analysis been performed appropriately and rigorously? 

Reviewer #1: Yes

Reviewer #2: No

3. Have the authors made all data underlying the findings in their manuscript fully available?

Reviewer #1: Yes

Reviewer #2: Yes

4. Is the manuscript presented in an intelligible fashion and written in standard English?

Reviewer #1: Yes

Reviewer #2: No

5. Review Comments to the Author

Reviewer #1: 1. Introduction would benefit from updated data on US HPV vaccine uptake. the citations presented are old.

2. In the results, you mention that the age of adolescents ranged 9-14, but that was the inclusion criteria, so it seems odd to present it this way. It is possible that ages may not have been included, but that would be a more appropriate way to report - e.g. "All ages in the eligibility criteria (9-14) were represented by adolescents in the study"

3. The finding about mood is confusing - what is the model by which this very acute measure at the time of the study could impact vaccination status? Additionally for the final items included, can you comment more on the clinical significance of including them?

4. Were all of the themes similar between parents and adolescents? Or were some more common in one group than another?

5. The discusion is mostly a list of other papers, but does not have a clear synthesis of all of the prior data and your findings. it's presented more like a review than like a clear discussion.

6. Were there any a priori sample size estimates generated, since N was identified as a limitation?

Reviewer #2: The paper “Encouraging improvement in HPV vaccination coverage among adolescent girls in Kampala, Uganda” addressed the topic of completion of the full cycle of vaccination against HPV. This is an interesting topic and the evaluation of underlying reasons of completion/uncompletion is important. Nonetheless, the papers shows some inconsistencies and problems that are reported hereafter.

Abstract

1. The abstract would need including more information in order to make the approach and, therefore, the conclusion clearer.

Introduction

2. Introduction needs to be deeply rephrased in order to avoid redundancy (need for two doses) and provided with more reference and data to support the need to assess vaccine completion and understand its determinants. The rationale of the study is not well framed. Furthermore, some parts are vague or miss time reference (i.e., “Two strategies were proposed, and some training was conducted for some health care workers. Based on anecdotal report by Global Alliance for Vaccines and Immunization full country evaluations (GAVI FCE) report 2016 poor national vaccine uptake and completion has been noted [5]. Uptake of HPV-1 in Uganda is at 73.8% while that of the second dose (HPV-2) at 17.3%”).

Methods

3. Authors quoted “The target population of the study was all girls aged 9-14 years in Kampala, Uganda and included all girls aged 9-14 years attending the adolescent clinic and girls who received the first dose at-least in the last 6 months in the clinic”: the time reference is not clear as the Authors further reported that “The principal investigator screened all files in the adolescent clinic from November 2015 to July 2017”.

4. Another point that needs to be addressed is the following: “Those who received the second dose 5-6 months after the 1st dose were then selected.“ This is not true because the study population was represented by girls receiving the first dose. The vaccine schedule completion within 5-6 months was the primary endpoint of the study.

5. Authors quoted “Incidence risk ratios for the factors were calculated with their 95% confidence intervals.”. Which variables did the Authors consider for this analysis? Statistical methods for the bivariate and multivariable analysis should be described.

6. The description of qualitative part should be made more fluent, avoiding redundant information (number of focus group performed, composition of groups, etc).

Results

7. Please provide the readers with a caption explaining the variable “class” and illustrate how school performance and mood were evaluated first and then extracted.

8. Table 3 should be made more readable as the name of variables are undistinguishable from categories and each variable should be provided with the number of observations with available information.

9. The results on the relationship between schooling status and HPV-2 completion were not significant. Please revise the text accordingly.

10. Results also includes some methodological aspects (variables entered in the multivariable analysis, evaluation of confounding). All these elements should be moved in methods. Furthermore, please keep in mind that because of the nature of the study no causal inference can be done. Indeed, why did the authors pay attention to confounding?

11. Motivations discussed under “Health Seeking Behaviors” seem to fall in peer influence.

12. In the reporting of qualitative data Authors quoted “Quantitative data indicates that majority of the adolescents (63%) attended the clinic for vaccination”. What does it refer to?

Discussion

13. Authors quoted “National completion levels for 2016, HPV-1 and HPV-2 was at 83% and 22% respectively while in 2017 HPV-1 was at 72% and 28% [5, 11]. The difference could be explained by the different models used in delivery of the vaccine i.e., school-based and age-based strategies, lack of follow-up for the second doses, inadequate information and guidance on eligibility criteria”. Please, provide more information about the calculation of these national coverage data.

14. There is an extensive discussion on the results coming from the qualitative analysis, but I am concerned about the fact that there was not any attempt to quantify themes and sub-themes recurrence among different groups. This is a major problem that prevents making final conclusions on the association between specific factors and completion. I would invite Authors to address this problem.

15. The study population in my opinion might be affected by selection bias and generalizability of results should be discussed further.

6. PLOS authors have the option to publish the peer review history of their article (what does this mean?). If published, this will include your full peer review and any attached files.

Reviewer #1: No

Reviewer #2: No

---

## [Author Response · Author response to Decision Letter 0]

30 Oct 2021

17th September 2021

The Editor,

Plos One 

Dear Sir/Madam,

Re: Revised Manuscript Submission 

Please find enclosed a revised manuscript entitled ‘Encouraging improvement in HPV vaccination coverage among adolescent girls in Kampala, Uganda’ We have responded to the comments of reviewers as follows: 

Reviewer #1: 

1. Introduction would benefit from updated data on US HPV vaccine uptake. the citations presented are old.

Done, included over 10 new updated references 

2. In the results, you mention that the age of adolescents ranged 9-14, but that was the inclusion criteria, so it seems odd to present it this way. It is possible that ages may not have been included, but that would be a more appropriate way to report - e.g. "All ages in the eligibility criteria (9-14) were represented by adolescents in the study"

Done, recommended change included

3. The finding about mood is confusing - what is the model by which this very acute measure at the time of the study could impact vaccination status? Additionally, for the final items included, can you comment more on the clinical significance of including them?

Done, explanation and reorganization included

4. Were all of the themes similar between parents and adolescents? Or were some more common in one group than another?

Discussed in text, and improved

5. The discusion is mostly a list of other papers, but does not have a clear synthesis of all of the prior data and your findings. it's presented more like a review than like a clear discussion.

Done, discussion fully redone with new updated citations 

6. Were there any a priori sample size estimates generated, since N was identified as a limitation?

Sample calculations done, but decision made to review all folders available this being a retrospective study and not many patients in the clinic.

Reviewer #2: 

The paper “Encouraging improvement in HPV vaccination coverage among adolescent girls in Kampala, Uganda” addressed the topic of completion of the full cycle of vaccination against HPV. This is an interesting topic and the evaluation of underlying reasons of completion/uncompletion is important. Nonetheless, the papers shows some inconsistencies and problems that are reported hereafter.

Abstract

1. The abstract would need including more information in order to make the approach and, therefore, the conclusion clearer.

Done, included

Introduction

2. Introduction needs to be deeply rephrased in order to avoid redundancy (need for two doses) and provided with more reference and data to support the need to assess vaccine completion and understand its determinants. The rationale of the study is not well framed. Furthermore, some parts are vague or miss time reference (i.e., “Two strategies were proposed, and some training was conducted for some health care workers. Based on anecdotal report by Global Alliance for Vaccines and Immunization full country evaluations (GAVI FCE) report 2016 poor national vaccine uptake and completion has been noted [5]. Uptake of HPV-1 in Uganda is at 73.8% while that of the second dose (HPV-2) at 17.3%”).

Introduction has been extensively redone, new references included, and the recommended changes added

Methods

3. Authors quoted “The target population of the study was all girls aged 9-14 years in Kampala, Uganda and included all girls aged 9-14 years attending the adolescent clinic and girls who received the first dose at-least in the last 6 months in the clinic”: the time reference is not clear as the Authors further reported that “The principal investigator screened all files in the adolescent clinic from November 2015 to July 2017”.

Done, these corrections have been added in the extensive review of the paper to reflect reviewer recommendations 

4. Another point that needs to be addressed is the following: “Those who received the second dose 5-6 months after the 1st dose were then selected.“ This is not true because the study population was represented by girls receiving the first dose. The vaccine schedule completion within 5-6 months was the primary endpoint of the study.

Done, this error has been corrected in text

5. Authors quoted “Incidence risk ratios for the factors were calculated with their 95% confidence intervals.”. Which variables did the Authors consider for this analysis? Statistical methods for the bivariate and multivariable analysis should be described.

Done, this error has been corrected in text

6. The description of qualitative part should be made more fluent, avoiding redundant information (number of focus group performed, composition of groups, etc).

Done, most repetitive text has been redacted 

Results

7. Please provide the readers with a caption explaining the variable “class” and illustrate how school performance and mood were evaluated first and then extracted.

Done, explanation provided and this aspect discussed in the discussion section 

8. Table 3 should be made more readable as the name of variables are undistinguishable from categories and each variable should be provided with the number of observations with available information.

Done, this has been corrected, table moved to supplementary documents 

9. The results on the relationship between schooling status and HPV-2 completion were not significant. Please revise the text accordingly.

Done, this error has been corrected in text

10. Results also includes some methodological aspects (variables entered in the multivariable analysis, evaluation of confounding). All these elements should be moved in methods. Furthermore, please keep in mind that because of the nature of the study no causal inference can be done. Indeed, why did the authors pay attention to confounding?

Done, this error has been corrected in text

11. Motivations discussed under “Health Seeking Behaviors” seem to fall in peer influence.

Done, this error has been corrected in text

12. In the reporting of qualitative data Authors quoted “Quantitative data indicates that majority of the adolescents (63%) attended the clinic for vaccination”. What does it refer to?

Done, this error has been corrected in text

Discussion

13. Authors quoted “National completion levels for 2016, HPV-1 and HPV-2 was at 83% and 22% respectively while in 2017 HPV-1 was at 72% and 28% [5, 11]. The difference could be explained by the different models used in delivery of the vaccine i.e., school-based and age-based strategies, lack of follow-up for the second doses, inadequate information and guidance on eligibility criteria”. Please, provide more information about the calculation of these national coverage data.

Done, the discussion has been extensively redone, corrections have been made, references updated to reflect the comments and guidance of the reviewers

14. There is an extensive discussion on the results coming from the qualitative analysis, but I am concerned about the fact that there was not any attempt to quantify themes and sub-themes recurrence among different groups. This is a major problem that prevents making final conclusions on the association between specific factors and completion. I would invite Authors to address this problem.

Efforts have been made to address this matter through the extensive revision of the discussion to reflect the comments and guidance of the reviewers. We are ready to modify this more to the satisfaction of the reviewers 

15. The study population in my opinion might be affected by selection bias and generalizability of results should be discussed further.

This limitation has been acknowledged, especially bearing in mind the retrospective nature of the study 

Please consider this article for publication in Plos One.

Yours sincerely,

Dr. Ombeva O. Malande.

MB.ChB, M.Med(Paed), FCPaed-Cert ID, MPhil(Paed ID),

Corresponding Author

Vaccinology & Paediatric Infectious Diseases Specialist,

Director, East Africa Centre for Vaccines and Immunization (ECAVI),

Lecturer Makerere/ Egerton University

---

## [Decision Letter · Decision Letter 1]

25 Jan 2022

PONE-D-21-09701R1

Encouraging improvement in HPV vaccination coverage among adolescent girls in Kampala, Uganda

PLOS ONE

Dear Dr. Malande,

Thank you for submitting your manuscript to PLOS ONE. After careful consideration, we feel that it has merit but does not fully meet PLOS ONE’s publication criteria as it currently stands. Therefore, we invite you to submit a revised version of the manuscript that addresses the points raised during the review process.

Some suggestions for modifications of the manuscript are listed in the Review 3 and include: the abstract, as well as some parts of the manuscript, e.g. Introduction, Methods, Discussion, Conclusion and some Tables. 

Furthermore,  your article will need a thorough review for readability and grammar. I would suggest the writing to be proofread by a native English speaker.

We look forward to receiving your revised manuscript.

Kind regards,

Prof. Maria Gańczak

Academic Editor

PLOS ONE

Journal Requirements:

Reviewers' comments:

Reviewer's Responses to Questions

**Comments to the Author**

1. If the authors have adequately addressed your comments raised in a previous round of review and you feel that this manuscript is now acceptable for publication, you may indicate that here to bypass the “Comments to the Author” section, enter your conflict of interest statement in the “Confidential to Editor” section, and submit your "Accept" recommendation.

Reviewer #1: All comments have been addressed

Reviewer #3: (No Response)

2. Is the manuscript technically sound, and do the data support the conclusions?

Reviewer #1: Yes

Reviewer #3: Yes

3. Has the statistical analysis been performed appropriately and rigorously? 

Reviewer #1: Yes

Reviewer #3: I Don't Know

4. Have the authors made all data underlying the findings in their manuscript fully available?

Reviewer #1: Yes

Reviewer #3: Yes

5. Is the manuscript presented in an intelligible fashion and written in standard English?

Reviewer #1: Yes

Reviewer #3: No

6. Review Comments to the Author

Reviewer #1: All prior comments have been sufficiently addressed. No further feedback for the authors to considers.

Reviewer #3: (No Response)

7. PLOS authors have the option to publish the peer review history of their article (what does this mean?). If published, this will include your full peer review and any attached files.

Reviewer #1: No

Reviewer #3: No

---

## [Author Response · Author response to Decision Letter 1]

27 Apr 2022

10th March 2022

Re: Response to Reviewers - manuscript entitled ‘Encouraging improvement in HPV vaccination coverage among adolescent girls in Kampala, Uganda’ We have responded to the comments of reviewers as follows: 

Abstract

Methods – Change data was to “data were” - DONE

Conclusion – what are the implications of your findings? What strategies or interventions could help achieve higher rates? - DONE

Introduction, p. 3

2nd paragraph, 

• Not certain why US rates are presented. If the authors want to draw parallels to Uganda then recent rates of vaccinations for girls should be presented there as well – CORRECTED 

• word missing from statement: There has been poor HPV vaccine uptake since its introduction into the national immunization program of . – CORRECTED

last paragraph

• what strategies or interventions did your ministry employ? Do they correspond with WHO guidelines or Community Guide to Preventive Services to increase vaccination rates? QUESTION HAS BEEN RESPONED TO IN TEXT.

p. 4, first paragraph

• needs a period at end of first sentence – CORRECTED

Methods

• What type of mixed methods design was this study? (concurrent, sequential). Please describe it in the methods. DONE

• Please describe if the vaccines are free of charge. This may be implicit but please make it explicit in this statement, if this is true. All vaccinations are supplied through the Ministry of Health’s EPI program DONE

P. 5, study procedures

• Change sentence to say “data… were”; need to change that for the rest of the paper as well DONE

• What was the multivariate analysis? Need to say the tests in methods. CORRECTED

• Last paragraph. Last sentence is a run on sentence; add “and” or “;” or make 2 sentences DONE

p. 6, first paragraph

• would recommend describing analyses and citing a reference to match your analyses (deductive based on interview guide, thematic? DONE

• Indicate initials from co author lists of the coders of the transcript or who did the analyses DONE

P. 7, ethical consideration

Last sentence, this is not needed: The findings are presented in this manuscript and in the supporting information files provided RETAINED, BECAUSE IT IS AN EDITORIAL REQUIREMENT FOR PLOS ONE 

Results, p. 8

• First paragraph, first sentence is a run on sentence NOTED, CORRECTED 

• Need to define what you are considering early vs late completion. Put in into the methods or in parentheses here DONE

• You may want to add other characteristics of the included girls into this paragraph DONE

Table 2

Suggest putting the N into the Title DONE

p. 10,

first sentence, would move Table 3 to the end of the sentence. The factors associated with timely completion of HPV-2 at bi-variable analysis were (Table 3) DONE

TABLE 3

Would put the key variables in bold like you did for Table 2 DONE

TABLE 5, PAGE 13

Could you indicate if the key themes arose from the FGD with the girls, parents or both. That would give a better sense of the strength of the themes DONE

p. 16,

change One went ahead and said: to “One commented” CORRECTED 

p. 19

• could remove 2nd sentence and put that under methods for what was consider timely: Completion at 6 months was considered timely. DONE

• what were your goal for vaccination of girls in Uganda or by WHO; could present that in the sentence: this figure still fell below the recommended national average for countries by WHO. DONE

• you discuss that facilities do not have strategies for reminders; what are the implications for this in terms of practice? THIS HAS BEEN EXPLAINED FURTHER

• Since your data may be difficult to compare to other countries, could you speak to other vaccinations for children and how this vaccination rate is and what parallels could be drawn? THIS HAS BEEN EXPLAINED FURTHER

• In the Discussion, could you describe any future research you would seek after the conduct of this study? THIS HAS BEEN EXPLAINED FURTHER

Conclusion

• First sentence; data do not need the parentheses CORRECTED 

• Perhaps add a sentence for what practice implications would increase the compliance with vaccine completion DONE

---

## [Editor Report · Decision Letter 2]

26 May 2022

Encouraging improvement in HPV vaccination coverage among adolescent girls in Kampala, Uganda

PONE-D-21-09701R2

Dear Dr. Oliver Ombeva Malanda,

We’re pleased to inform you that your manuscript has been judged scientifically suitable for publication and will be formally accepted for publication once it meets all outstanding technical requirements.

Kind regards,

Prof. Maria Gańczak

Section Editor

PLOS ONE
---

## [Editor Report · Acceptance letter]

31 May 2022

PONE-D-21-09701R2 

Encouraging improvement in HPV vaccination coverage among adolescent girls in Kampala, Uganda 

Dear Dr. Malande:

I'm pleased to inform you that your manuscript has been deemed suitable for publication in PLOS ONE. Congratulations! Your manuscript is now with our production department. 

Kind regards, 

on behalf of

Prof. Maria Gańczak 

Section Editor

PLOS ONE